# The Urban Mirror of the Socioeconomic Transformations in Spain

Josefina Domínguez-Mujica 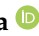

Department of Geography, University of Las Palmas de Gran Canaria, 35003 Las Palmas, Spain; josefina.dominguezmujica@ulpgc.es

**Abstract:** This study offers an interpretation of the most significant characteristics of Spanish cities in the post-Fordist capitalist era, as a mirror of the economic and social transformations that have led to them, differentiating: (i) the stage of economic expansion at the turn of the century; (ii) the stage of the economic crisis from 2008 onwards; and (iii) the uncertain times we face for the future. Therefore, the aim of this paper is to identify the economic, housing, and political factors conditioning this evolution according to the processes of capitalist accumulation, dispossession, and repossession, and how they shape the social and urban configuration of Spanish cities. A careful selection of urban and economic indicators, its mapping, as well as an in-depth bibliographical review lead to this commentary and make it possible to identify urban developments in Spain in the light of the economic and social transformations of post-industrial capitalism.

**Keywords:** Spanish cities; economic crisis; financialization; gentrification; housing crisis; post-Fordist capitalism; property bubble; property repossession; touristification; pandemic

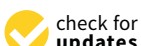



## 1. Introduction

On 13 July 1986, an article entitled "El niño 5.000 millones será pobre" ("The 5000 million child will be poor") was published in the *El País* newspaper [1], an article that I cut out and photocopied, using the teaching resources of the time, to discuss in class. It said that the baby who officially came into this world on 7 July of that year could be Chinese or Indian and that it was possible that he or she would be born in a village, but it was also likely that he or she would end up leaving the countryside to join those marginalized crowds that settled in the outskirts of the big cities.

Twenty years later, Mike Davis in his book *Planet of Slums* [2] stated that in 2006 a young man would flee his village in West Java to go to the bright lights of Jakarta and that a Peruvian farmer would migrate with his impoverished family from his hometown to one of the countless "pueblos jóvenes" in Lima. These were metaphors for the unstoppable process of the urbanization of the Earth, which, according to Kate D. Derickson, has phagocytized the city and the countryside and involves the complete urbanization of society [3,4]. Thus, although cities only occupy 2% of the planet's surface, they concentrate more than half the world's population, consume 75% of the energy, and produce 80% of the GDP.

This socio-spatial transformation has taken place in the last 30 to 40 years of our history because of an economic, political, and cultural logic that has determined the global expansion of the productive forces and the subsequent annihilation of space by time and of time by space. As we all know, this is a phase of late and post-industrial capitalism that has generated greater urban complexity and has been at the center of innumerable analyses, due to the increase in the quality and quantity of interactions between cities and of the internal relations that take place in the urban fabric itself.

The city has thus responded to globalization, an ambiguous and contradictory process that, while promoting interconnections and convergences, shortens distances and opens up meetings between peoples and social groups, accentuates distances and inequalities. Despite the differences among the various cities on Earth, the qualities inherent in the urban fabric, at the pace of globalization, are the result of the interaction of two forces. On the

one hand, there is the concentration of capital, labor, and culture in the city, and, on the other hand, the radical transformation of its economic base, through the passage from a Keynesian and Fordist system of mass production and consumption to a post-Fordist system of flexible, information-intensive industrialization associated with the vertical disintegration of the production process. This has led to a progressive relocation of some of the industries and the consequent tertiarization of the urban economies, with services now accounting for more than three-quarters or four-fifths of their employment.

The unprecedented concentration of power, wealth, and knowledge, typical of the city at this stage, also brings with it unprecedented social exclusion. Metropolises, technopolises, and globalized regions exchange economic information, technological and human capital resources, and, at the same time, disconnected territories are weakened and dependent, if not excluded from the process of economic development. As a result, vulnerable spaces are consolidated in the city itself, those in which there is increasing risk, more precarious employment, greater insecurity, and a scarce capacity for decision-making, with dynamics of residual characteristics. Parallel to these processes, in the words of Nogué and Romero [5], urban society is configured by an enormous number of groups, differentiated by their interests; social, ethnic, economic, political, sexual, identity, and generational components; lifestyles; access to resources, power, well-being, time, etc., because the city is consolidated as a powerful machine of differentiation and separation, of marginalization and exclusion.

This context is validated by the opinions of such well-known authors as Thomas Piketty [6], Amartya Sen [7], José Antonio Ocampo [8] and the United Nations *Human Development Report* [9], when they pointed out that inequality among nations has been reduced during the third globalization, but inequality has generated within a vulnerable citizen's state of mind "persistent inequalities." These inequalities of income and wealth that define an urban landscape of unbalanced development through increased social polarization and residential segregation do not disappear. Thus, the late-capitalist metropolises, or what some have defined as the dual city, are evolving at a simultaneous pace with the tendencies of capitalism towards equalization and differentiation, and towards devaluation and revaluation, according to the late geographer Neil Smith [10].

In this scenario, the processes of transformation become difficult to comprehend, and it becomes complicated to put forward a discourse on the immediate future. Many authors agree that, in the coming years, incessant urban growth will be accompanied by greater diversity, accentuating complexity, with diffuse, concentrated, and mixed models of urbanization that face the challenge of sustainability. Therefore, before our eyes the urban reality shows that we are its discontinuous and conflictive agents, and times are accelerating at the pace of information and communication technologies, which makes it difficult to understand new situations. As Secchi pointed out [11], in times of transition we are not able to use simple words to speak about the city.

Against this backdrop, a specialist in population geography faces the boldness, rather than the challenge, of interpreting some of the most significant trends in Spanish urban development at this stage of post-Fordist capitalism, trying to differentiate the elements of economic expansion at the turn of the century, those of the economic crisis and of the times of uncertainty with which we look to the future. Consequently, the next sections will identify the economic, housing, and political factors conditioning this evolution because the aim of this paper is to offer a reflection on the transformations of Spanish cities in the light of the evolution of post-Fordist capitalism, characterized by the concatenated processes of accumulation, dispossession, and repossession. With this objective, a methodology based on an in-depth bibliographical review and a selection of different economic and housing indexes was used, providing a comprehensive interpretation of the recent Spanish urban evolution.

## 2. Materials and Methods

Starting from the premise that urbanization is a relevant expression of the economic, cultural, social, and political characteristics of a society [12], the theoretical reflection that I present was made after consulting many empirical works on recent trends in Spanish urban development. To do this, I proceeded as follows: on the one hand, I selected the books and articles published in scientific journals that were referenced on the Google Scholar portal according to the following topics: economic crisis; financialization; gentrification; housing crisis; post-Fordist capitalism; property bubble; property repossession; Spanish cities; touristification; and pandemic. On the other hand, I reviewed the contributions to the Urban Geography Colloquia promoted by the Association of Spanish Geographers, choosing those that offered an overview, or case studies carried out from a critical perspective. To all of them I added a few articles published in magazines.

The time horizon was 2008 onwards because, although the study deals with the characteristics of the immediately preceding period, I considered that it was necessary to use only the interpretations that were published from that date because they offered a critical perspective in the light of the events that took place after the decline and recession of the global economy and, particularly, of the Spanish economy. Only a few conceptual reflections of notable importance escaped this dating.

Once the collection of these publications was made, a database was created in which the articles were classified according to their chronology and subjects of study, having subsequently chosen those I considered most appropriate for the purposes of this commentary paper. Finally, to all this I added a selection of economic, housing, and property registration indicators, as well as of judicial procedures, to elaborate the figures. To them I added two conceptual schemes of the factors conditioning the economic housing expansion and the economic housing crisis.

## 3. Results

The conceptual approach to the most significant characteristics of Spanish cities in the post-Fordist capitalism era required consideration of three different stages: (i) the stage of economic expansion at the turn of the century, with the liberalization of mortgages and the land market; the securitization of loans and the expansion of credit; and the promotion of urban land and construction, as sources of funding for institutions; (ii) the stage of the economic crisis from 2008 onwards, with the loss of value of property, dissaving, reduction in income and capital and its consequent effects on dispossession; and (iii) the uncertain times we face for the future, with an incipient economic recovery, with processes of repossession, through the progressive elimination of toxic assets from the financial system and the appearance of new operators—investment funds that acquire, promote, and rehabilitate goods of an urban nature.

### 3.1. The Urban Configuration in Spain during the Economic Expansion at the Turn of the Century

After the developmentalism of the 1960s, and a long period of diffusion of urban dynamics from the largest cities to the smallest agglomerations, after the crisis of 1973 and later, different economic situations have followed one another in Spain that today can be interpreted in the light of the new tensions of an incipient globalization. This was the prelude to the intense economic growth that we witnessed at the turn of the century; a period that was impossible to dissociate from real estate growth itself and that gave rise to unprecedented urban expansion of a disorderly nature, with a notable increase in artificial surfaces and indiscriminate land consumption. Furthermore, it remained a common trend in southern Europe, where over-urbanization based on hyper-compact and deregulated expansion, real estate speculation, and informal economy were general factors of the cities' growth [13,14].

The increase in artificial land in Spain can be recognized on the map in Figure 1, which represents its variation between 2000 and 2006, according to data from the European CORINE (Coordination of Information on the Environment) Land Cover project. The GDP

Interannual Variation Rate (%) is also represented in the map, but there is not a clear positive correlation between these indicators by provinces, given that the artificial land's increase is also related to the investment in second houses, to foreign investment, and to a process of suburbanization that sometimes exceeds the provincial limits.

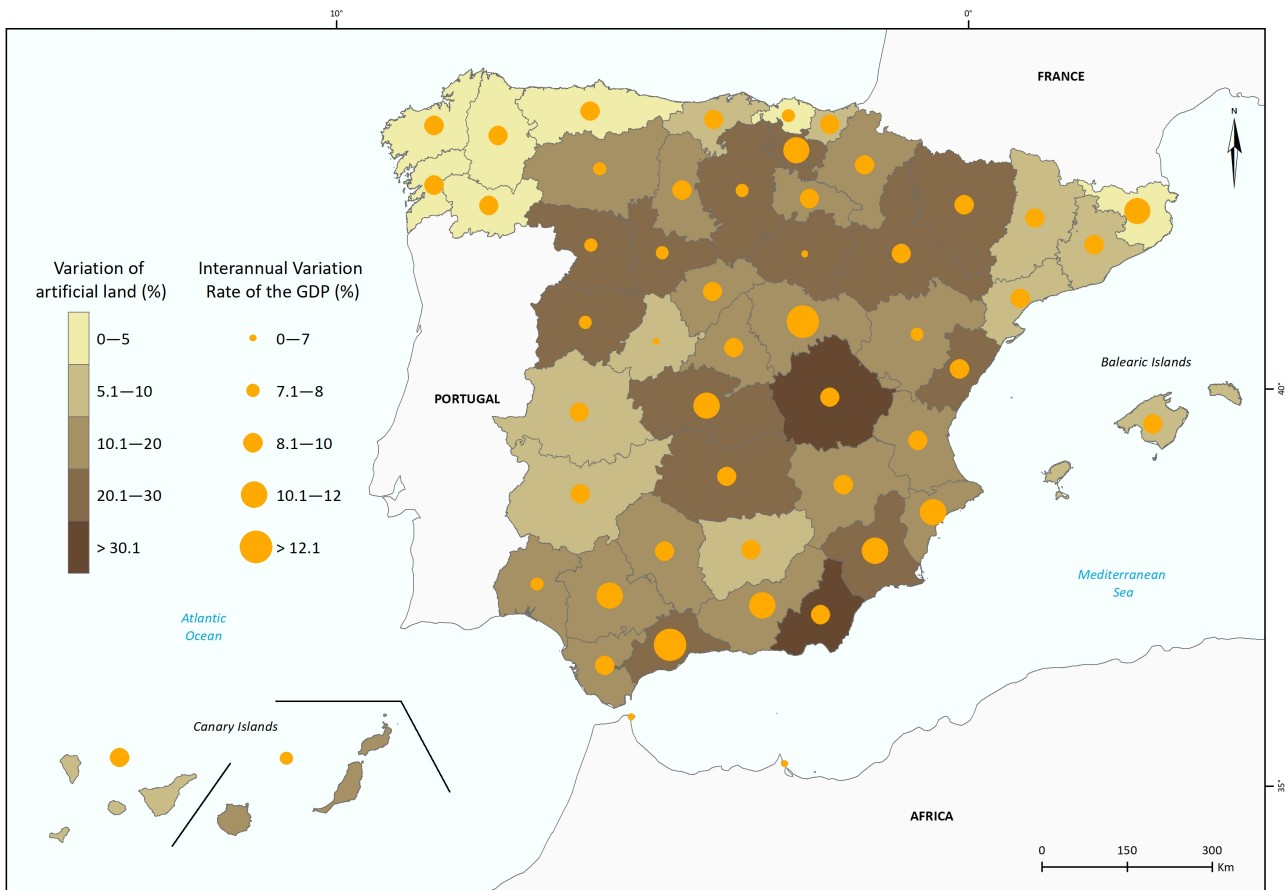

**Figure 1.** Variation of artificial land (2000–2006)% (source: Digital Atlas of Urban Areas [15]) and Interannual Variation Rate of the GDP (2000–2006)% (source: National Accounts of Spain: Main Aggregates (QNA). Spanish Institute of Statistics (INE) [16]). Author's elaboration.

Consequently, the city at this stage was forged as an image of a real estate bubble, as a result of the business expectations of landowners, developers, and builders and their collusion with the financial system and the political–administrative system, demonstrating what David Harvey [17] defended: that urbanization has been one of the means of absorbing surplus capital and labor throughout the history of capitalism, and that land is not a commodity in itself, but a fictitious form of capital, derived from expectations of future income.

It was the year 2000 when the housing sector began to evolve wildly. Prices were rising by 17% a year with very low inflation, which implied high growth in real terms. Until 2006, an average of 600,000 homes were started each year, a figure higher than that of Germany, Italy, and France combined, according to data from the current Ministry of Transport, Mobility, and the Urban Agenda (Ministerio de Transportes, Movilidad y Agenda Urbana) [18]. Cranes were common in the landscape, almost an element of national pride, in the words of a columnist from *El País* who referred to this situation [19]. However, this increase in housing construction was not accompanied by a reduction in its price; on the contrary, it did not respond to the demographic growth of cities and had disastrous consequences on the indebtedness of households in mortgage loans, a debt that increased by 738% between 1996 and 2007 [20].

This scenario was made possible by the fact that, since the mid-1980s, Spanish policy had been committed to the secondary accumulation circuit and to the financialization of the economy as the main mechanism for growth, taking advantage of the benefits of Spain's recent integration into the European Community (Figure 2).

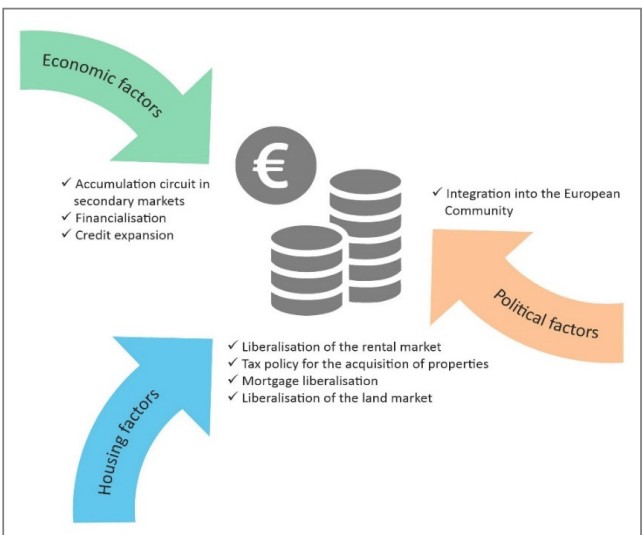

**Figure 2.** Factors conditioning the economic housing expansion in Spain (1998–2008). Source: author's elaboration.

### 3.1.1. What Urban Image Does This Period of Prosperity Project?

As opposed to the concentrated model of developmentalism with an over-dimensioned and under-utilized housing stock, the image of a diffuse city with important suburban growth in peripheries where new developments are contrasted with an important consumption of land is opposed to the so-called gothic suburbs, desolate old suburbs, in which less favored social classes reside. This deconcentration, which characterizes this stage, coexists with the evolution of urban centers that are still degraded and aging, affected by the first processes of gentrification, that is, by an incipient transformation of the working-class neighborhoods or empty spaces in the central city into areas for residential, commercial, or service use by the middle class, all of which implies the displacement of low-income social groups.

### 3.1.2. And What Is the Configuration of Urban Society in These Years of Economic Expansion?

Urban society is one of cultural heterogeneity, which results from the increase in transnational migrations or, as some authors have also defined, from labor migration, that is, the city in which the spaces of flow emerge, in the words of Castells [21]. In addition to this, there is a new social awareness, with the development of values that have contributed to define a more heterogeneous city in terms of ethnicity and culture, with mixed, in-between, border, and contact spaces.

To the ethnic communities and the external image they project, in the use of public spaces, in small businesses, or in the offer of certain services, new social and identity values are added, whose recognition is demanded by homosexuals, transsexuals, and bisexuals in well-defined urban spaces, spaces that they shape. These practices allow us to understand urban entities in a different way, as the Spanish city of this period begins to manifest itself as a city sensitive to the subject and to the personal experiences of women and men; that is, they allow us to make a geography of everyday life, to rescue movement, in the words of David Ley [22] (p. 162): "transnational spaces, yes, but still everyday lives."

These transformations coincide with a change in the labor market pyramid, with visible contrasts between the base of workers with lower incomes and in a more precarious situation, and the managerial–executive social class. This division, although reminis-

cent of the proletariat–bourgeoisie, is more complex, blurred in its limits, with greater overlaps, and less predictable from a political point of view, due to social fragmentation, ideological plurality, and the proliferation of interests found.

### 3.2. The Impact of the 2008 Global Economic Crisis on the Spanish City

Periodic economic crises and their various manifestations are inherent in the very logic of the capitalist system, since the basic tendency to over-accumulation generates, episodically, surpluses of capital, scarcity of investment opportunities, falling rates of profit, and a lack of effective demand in the market. Consequently, according to Harvey [23], they apply a certain order and rationality to capitalist economic development, since they expand the productive capacity and reintroduce the conditions for a new accumulation, although, at the same time, they lead to dramatic social consequences: financial collapse, forced devaluation of capital assets and personal savings, inflation, a fall in real wages, and unemployment.

The economic crisis that occurred in the United States in 2007 and spread globally due to the bursting of a large financial bubble was a response to this profile. The latter had been forged by the practice by US banks of issuing real estate bonds that offered high profits and low risk through deregulation mechanisms, a procedure that was copied by many other countries, such as Spain. To keep the capital flow of these bonds constant, mortgage loans were granted in bulk, the so-called "subprime" mortgages, but when investors demanded payment of these bonds, interest rates rose because nobody knew what the packages of securities contained. The banks were unable to respond to this demand and a liquidity crisis, a credit crisis, and, as a result, an employment crisis ensued.

In southern Europe, austerity was marked by a shift toward a starker version of neoliberal doctrine, but the crisis did not lead to the reforms expected, at least not in the field of housing [24]. In the particular case of Spain, the system of accumulation of surpluses in the secondary circuit, which had been based on the collaboration of the building–real estate sector, the financial system, and the political–administrative system, turned into over-accumulation when financing contracted in 2008 in the face of a lower outlook for performance, with the consequences of a decrease in demand, greater difficulties for promoters in selling and financing, and a complete contraction of the sector, which affected the economy as a whole [25] (Figure 3).

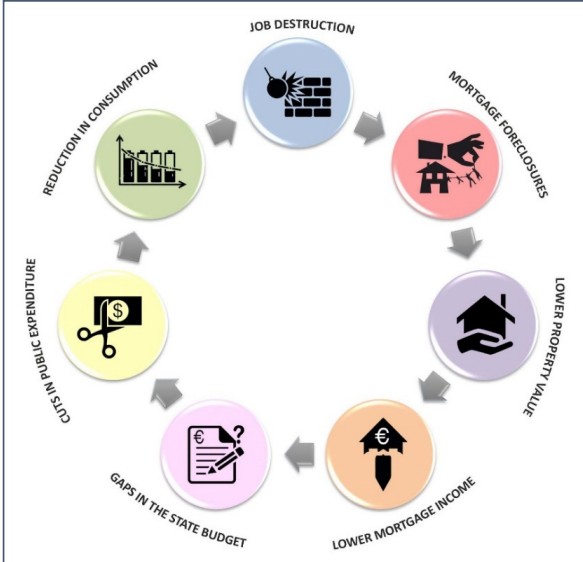

**Figure 3.** Factors conditioning the economic housing crisis in Spain (2008–2015). Source: author's elaboration.

A good example of the drop in the general housing price, by autonomous communities, in 20081Q–20141Q is shown in Figure 4. This dynamic is related to the fact that, in those autonomous communities where the prices were highest at the beginning of the real estate bubble bust, the fall was greater [26].

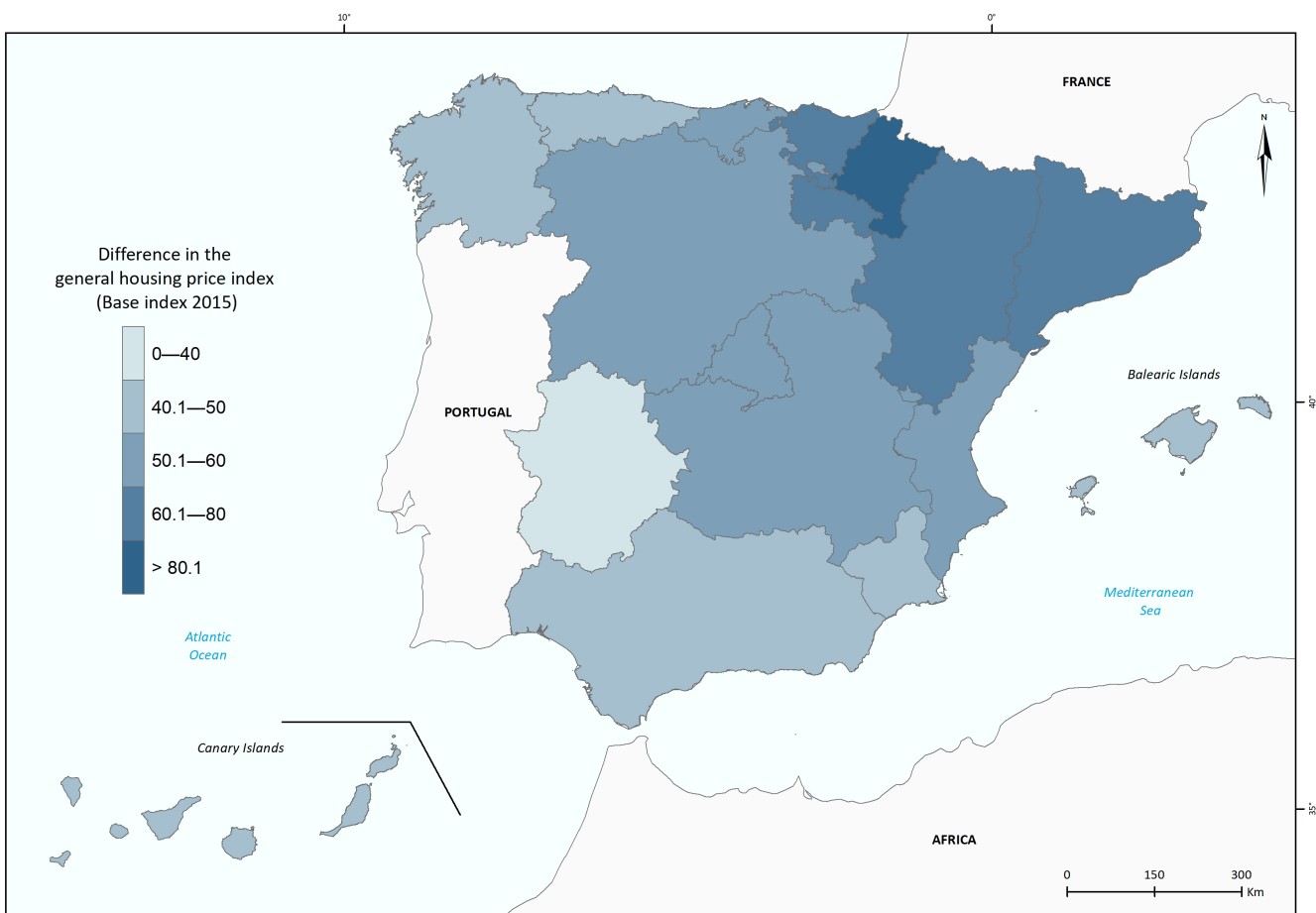

**Figure 4.** Difference in the general housing price index (Base index 2015) (20081Q–20141Q). Source: Housing Price Index (HPI), National Statistics Institute, Spain (INE) [27]. Author's elaboration.

Otherwise, the economic crisis and rising unemployment led to a dramatic increase in mortgage defaults by buyers. According to Méndez [28], the growing socio-professional precariousness spread to professional groups, to the middle classes, and to territories that seemed immune. Wage devaluation polarized the labor pyramid and increased the distance between its extremes, and the mortgage-holders who could not meet their loans could not use the house as an asset to save the situation, either, because the market prices, which were logically falling, were lower than the mortgage amounts contracted.

This led to a rise in foreclosures and repossessions, although in many cases the assets did not allow the balance sheets of the lending institutions to be restored, given their lower value on the market. Likewise, the worsening economic situation of part of the population put rent payments at risk, so that evictions for non-payment of rent, another side of the same process, were also generated. In other words, according to Romero [29], the virtuous circle of the financial and real estate economy in the expansion stage soon turned into a vicious circle of loss of value, eviction, and reduction of income and capital, with the aggravating factor that an institutional system was not activated to alleviate its devastating social effects, either through the transfer of income or by making payment conditions more flexible.

### 3.2.1. How Did the Economic Crisis Transfer to the Urban Crisis?

The economic crisis became the urban crisis with the multiplication of ghost housing estates, which spread all over the map of Spain, and the bursting of the real estate bubble. The urbanistic excesses that accompanied it left traces that are difficult to erase: unfinished housing developments, almost completely unoccupied blocks of flats and, around them, the desolation of streets, streetlights, litter bins, and new, unopened playgrounds. In the words of Markel Redondo, the photographer who has traveled all over Spain taking snapshots of some of the many urbanizations and ghost complexes that the brick bubble left all over the country: "To be in those places is strange, you get the feeling of being the last human being on earth, contemplating the ruins" [30]. For the *Málaga Hoy* newspaper, these urbanistic excesses on the Costa del Sol left "a coastline of real estate corpses" [31].

As we all know, the Mediterranean was the area with the greatest volume of flats built, especially Murcia, the Valencian Community and Andalusia, which accumulated the newest empty flats in relation to the total housing stock, and which, in 2017, were still the autonomous communities with the greatest difference between completed homes and new home sales since 2004. In Spain as a whole, even though for several years more new housing was bought than was built, there were still almost half a million unsold homes in 2018, according to data from the Bank of Spain [32].

On the other hand, in Ireland and the USA, after the bursting of the real estate bubble, many buildings were demolished to reduce the supply and because conserving them was more expensive than tearing them down. In Spain, although experts believed that unfinished and badly situated works would be demolished, the Sociedad de Gestión de Activos Procedentes de la Reestructuración Bancaria (SAREB) (Company for the Management of Assets from Bank Restructuring) has not undertaken a clear demolition plan, despite allocating a significant amount to the maintenance of buildings and to works in progress in empty urbanizations. Thus, the ghost developments in the urban peripheries of many Spanish cities have been waiting for someone to decide if the cranes will return, showing the scars of the crisis following the real estate bubble.

### 3.2.2. What Are the Social Characteristics of the City during the Crisis?

I could say that the social characteristics of the city during the crisis are those of a living and committed city.

According to Edward Soja, urban tensions are among the most explosive in the world in social terms and pose the greatest political challenges [33]. This is how he identified the episodes of racial violence in the United States and they also fit in with the urban manifestations of the so-called "Arab Spring."

There is no doubt that these events are far from the conflict arising from the economic crisis in Spain and its urban repercussions. However, there is some truth in Soya's maxim if we take into account the emergence of certain movements of reaction from the so-called civil society to the evictions due to the non-payment of mortgage loans. The first Platform for People Affected by Mortgages was created in Barcelona in 2009 [34], since the legal framework had been designed to guarantee that banks would collect the debts, while leaving unprotected those people with mortgages who, for reasons such as unemployment or rising fees, were unable to pay.

The Platforms of People Affected by Mortgages, which define themselves as places of meeting, help, and action for those affected, as well as for people who are in solidarity with this reality, have multiplied throughout Spain and developed different campaigns. Among these the STOP EVICTION campaign stands out, in which they are calling for action because, according to their own declarations: "We will not allow any more evictions! We will not let the bank throw us out of our homes!" With a clear urban identity, the movement of the outraged or 15–M was also born, revealing Harvey's maxim that "there is something political in the city air struggling to be expressed" [35] (p. 117).

In this case, the citizens' movement was formed as a result of the demonstration on 15 May 2011, convened by various groups that promoted a series of peaceful protests with the intention of defending a more participatory democracy [36]. It was a movement that generated a public debate on the representativeness of political institutions, electoral rules, dation in payment, transparency in the remuneration of high officials and corruption.

Emerging from the impoverishment created by the economic crisis of the preceding years and from the low expectations of young people, the malaise expressed by its supporters found its main tools of expression in the Internet and social networks. These channels of communication guaranteed its echo throughout the world, with more or less massive demonstrations in London, Paris, Brussels, Rome, Lisbon, Washington, New York, Berlin, Frankfurt, Tel Aviv, Rabat, Wellington, Taipei, Seoul, Tokyo, etc., in a transnational process that was not alien to the activism that launched a criticism of political power and protested against the consequences of the functioning of markets and banks, against cuts, or against precariousness in employment.

*3.3. The Capitalist Restructuring of the Spanish City after the Economic Crisis*

The political reactions to the 2008 crisis, after an initial moment of denial of the situation, arose from 2010 onwards, when the first measures were taken, focusing exclusively on bank restructuring through the merger of savings banks and their conversion into private banks. Two years later, Bankia and Catalunya Bank were nationalized and SAREB was created, a company aimed at absorbing the toxic assets of the new banks [37]. In other words, through public debt, the losses of the financial institutions were transferred to a semi-public bank, guaranteeing the solvency of the companies through the financial rescue of their toxic assets, but without establishing measures that addressed the problem from the perspective of those affected. Therefore, the issue of abusive floor clauses in mortgages was not resolved; the dation in payment, i.e., the handing over of the mortgaged property in exchange for the cancellation of the debt, was ignored; and the daily drama of evictions, which affected a significant part of the population, was not addressed. For the government, the problem was reduced to the elimination of toxic assets from the financial system, ignoring the growing social unrest that was taking place, until a European ruling was issued in 2017 that forced banks to make the floor clauses totally retroactive.

To these trends we must add the fact that, as of 2013, the progressive sale of these assets began in addition to the recovery of real estate activity (investment, construction, sales, prices) and also the granting of mortgages, while the rental market was reactivated and a policy of attracting international investors began through the granting of tax benefits and the elimination of bureaucratic barriers [38]. New operators appeared, such as the SOCIMIs (Sociedades Anónimas Cotizadas de Inversión Inmobiliaria), which, as listed real estate investment companies, have as their main activity the acquisition, promotion, and rehabilitation of urban assets for lease in a context of im-proved profitability prospects. In addition, investment funds, the so-called vulture funds, took over part of the assets of SAREB and the other financial institutions at more than advantageous prices, through instruments such as the Real Estate Management Companies [39].

This internationalization of home ownership through a policy that ensured a favorable benefit for large international investors, powerful economic agents, and financial institutions was the final step in its financialization and is the necessary requirement for understanding that it has gone from a model of accumulation to one of accumulation by dispossession and subsequent repossession. In other words, the idea already expressed by Neil Smith that neoliberalism turns cities into centers of production for the global economy [40] increasingly eliminates the functions of social reproduction and uses them as a test bed for the new entrepreneurship; a facet of hardcore capitalism that accentuates the process of the commodification of the city is fulfilled.

Two other factors have added to this panorama of the internationalization of housing and are having a great impact on the property market and, therefore, on the restructuring

of the city after the crisis. I am referring to the so-called Golden Visa and the so-called collaborative or platform or network economy.

Law 14/2013, of 27 September, on the support for entrepreneurs and their internationalization, established a special residence visa for investors, commonly known as the Golden Visa, which allows any large investor in real estate (worth EUR 500,000 or more) to live for one year in the entire national territory and includes a work permit [41]. The Golden Visa also permits the holder to bring his or her family members to Spain.

Since this program was launched, the number of permits has been increasing year by year. In total, 8061 of these visas were granted in 2019 [42] and, according to data from the Ministry of Labour, Migrations and Social Security and the Ministry of Foreign Affairs, European Union and Cooperation, at the top of the list of foreigners who decide to invest in Spain and, in exchange, obtain a work and residence permit, are Chinese, Russian, Ukrainian, Iranian, American, Mexican, Venezuelan, Indian, Brazilian, etc., citizens who choose, preferably, properties in Madrid, Barcelona, and Málaga in absolute terms, and in the Mediterranean provinces and the two archipelagos in relative terms (Figure 5).

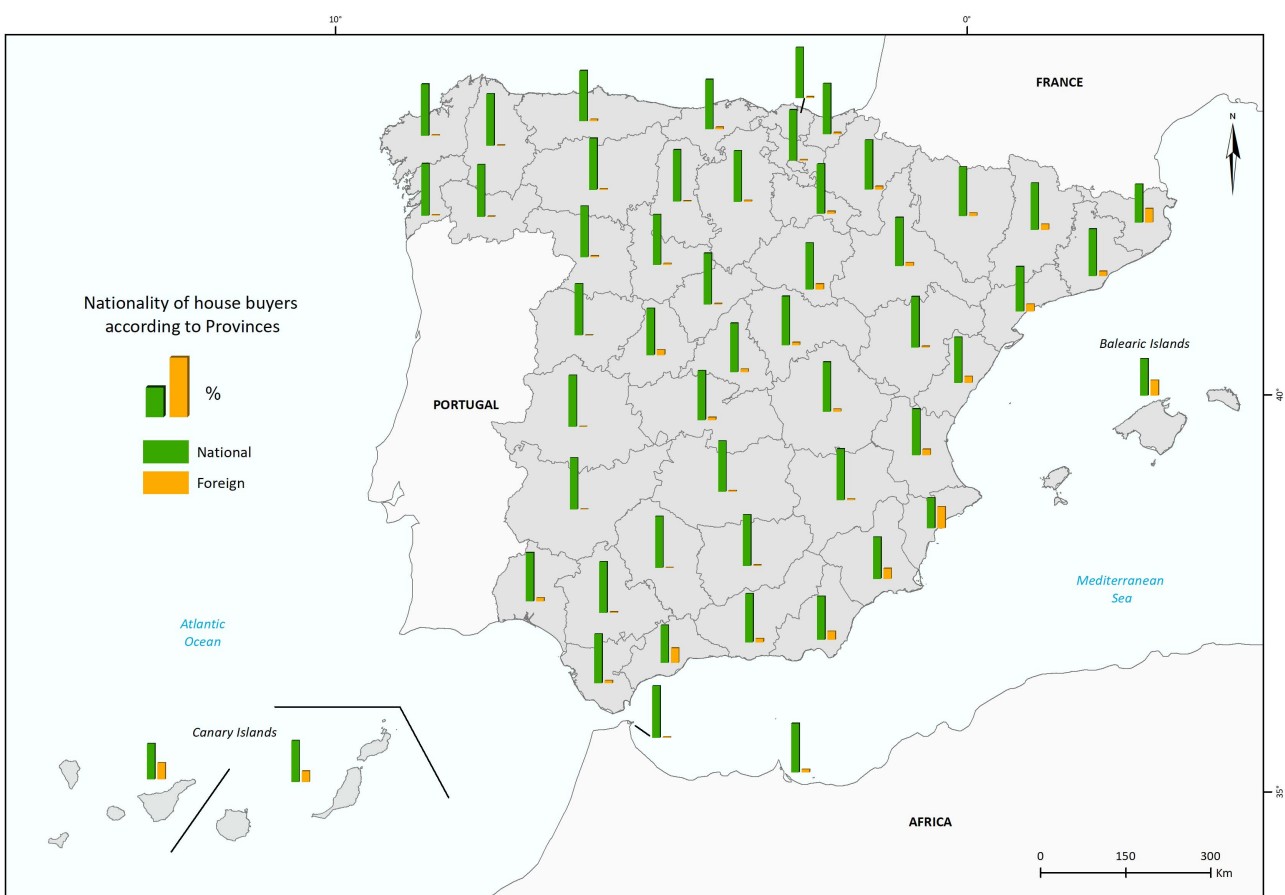

**Figure 5.** Housing buyers according to their nationality (%) by Spanish provinces in 2019. Source: Association of Property Registrars of Spain [43]. Author's elaboration.

Chinese investments are well-known in our country and are some of the most solid. So much so, that many real estate agencies have a unique division for this Asian country. In fact, until 2018, some Chinese citizens had managed to invest up to EUR 821 million in Spain in exchange for more than 1200 residence permits, a third of the total granted. They were followed by Russians and Ukrainians, two of the traditional nationalities who acquire, above all, homes on the Mediterranean coast.

Finally, the presence of large Venezuelan fortunes in the Spanish real estate market shot up in 2017 and 2018, due to the political and economic situation the Latin American country was going through, with purchases of luxury properties that are generally paid in cash and

reach up to EUR 30 million. Although these fortunes have always been interested in Spain, as evidenced by the fact that Novagalicia Banco was awarded by the FROB (Fondo de Reestructuración Ordenada Bancaria) to the Venezuelan group Banesco in 2013 [44], for the last three or four years there has been a sharp increase in Venezuelan buyers and investors, especially in Madrid's Golden Mile, in the heart of the Salamanca district. As a result of these real estate investments by non-EU foreigners and foreigners from European Union countries, Spain was leading the southern European real estate recuperation in 2017 [45]. Thus, Figure 6, which represents the evolution (by quarters) of the housing prices and of the housing acquisitions by foreigners between 2014 and 2019, allows us to appreciate that in the generalized increase in housing prices due to the Spanish economic recovery, the sustained evolution of foreign investment has also favorably contributed.

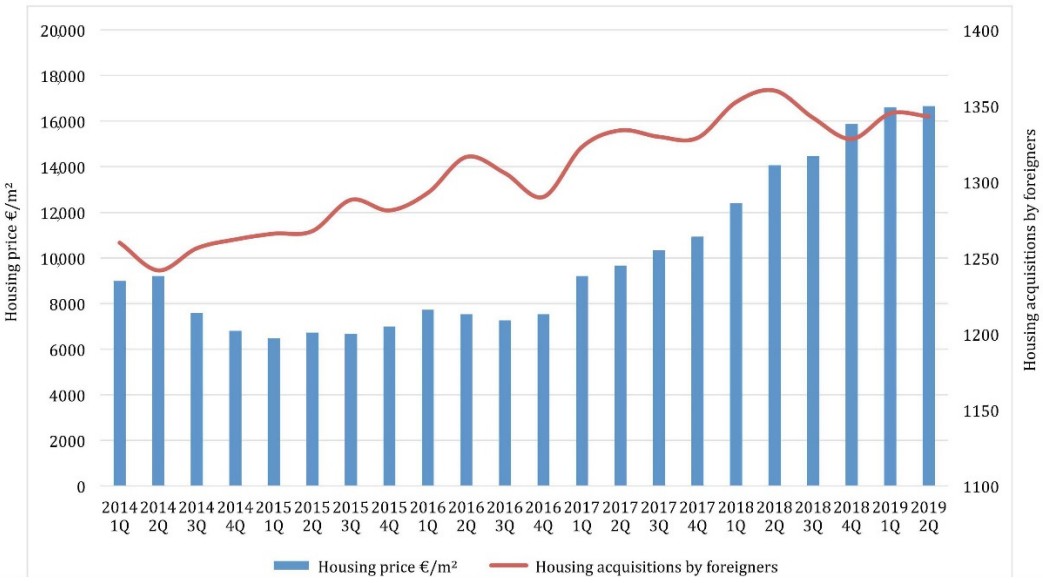

**Figure 6.** Evolution (by quarters) of the housing prices and of the housing acquisitions by foreigners. Sources: Tinsa, appraisal company approved by the Bank of Spain [46]. Association of Property Registrars. Real Estate Registry Statistics [43]. Author's elaboration.

The second factor I mentioned is that of the collaborative economy. As we all know, this is a new business model which, by means of Web platforms, allows services to be exchanged for money, taking a commission. The platform that best exemplifies this philosophy in relation to our interests is Airbnb. It was originally created in San Francisco, in 2008, to rent homes with an inflatable mattress and breakfast (air bed and breakfast) to guests who were passing through the city, which gave the company its name.

The development of this platform, as a start-up, attracted the attention of a business financier and several investment funds, which injected large sums of capital into it. As a result, the initiative landed in every tourist place on Earth and the company became one of the eight global giants in technology, so other similar companies supported the initiative, such as Home Away or Booking Home.

The rental of housing for tourist uses as an activity of this cool economy has had a negative impact on the housing market, as it remove from the offer many of the properties that were intended for permanent residential use, since it is a more lucrative business than the traditional rental one. This, in turn, implies processes of displacement and social elitization in city centers as well as an increase in second homes, as opposed to traditional neighborhoods. Consequently, while the number of tourist dwellings has increased, especially in the coastal provinces and in the two archipelagos, there has also been a high number of evictions, as depicted in Figure 7, so that both variables show a non-parametric correlation (Spearman = 0.785).

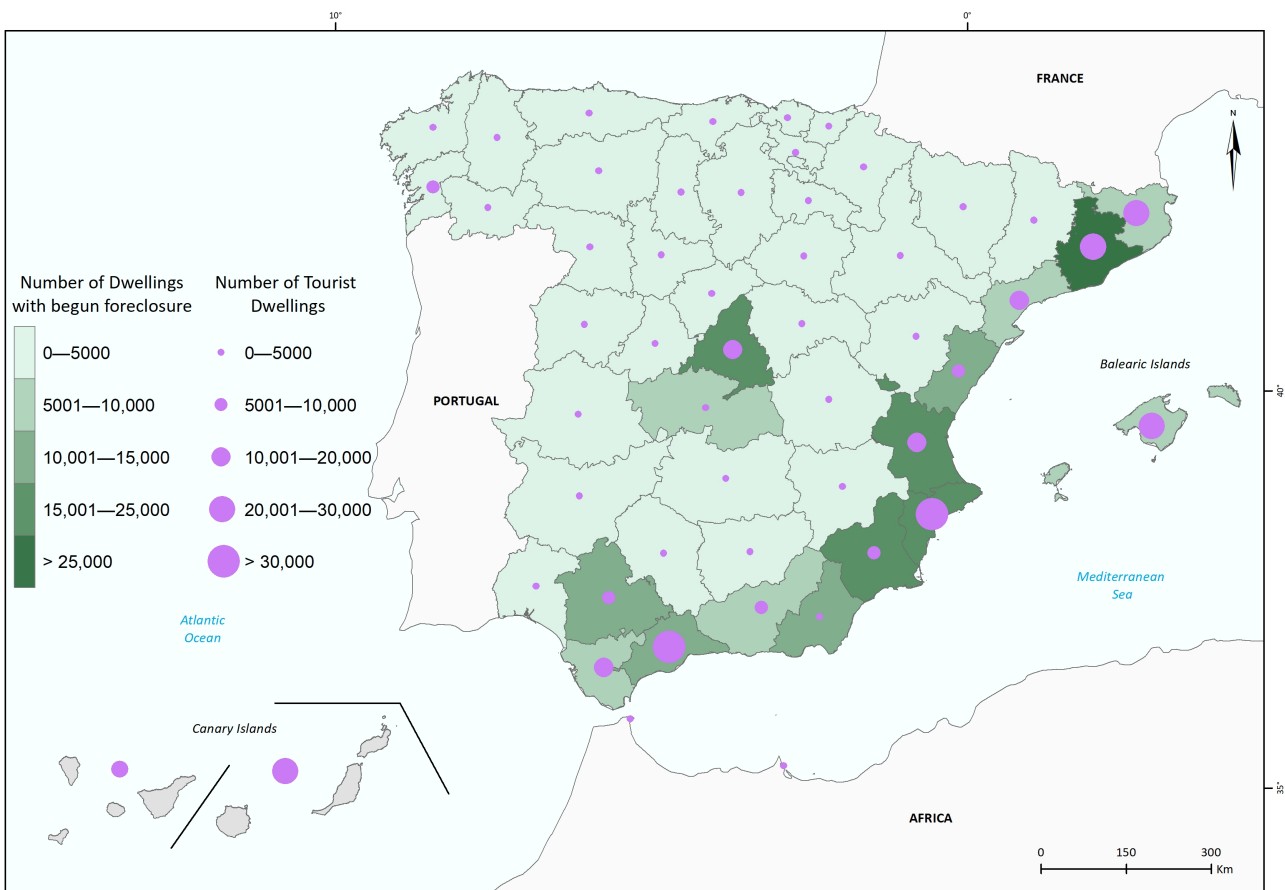

**Figure 7.** Tourist dwellings by province in absolute figures in 2020, and number of dwellings with foreclosure begun by province, 2014–2019. Source: Measurement of the Number of Tourist Dwellings by Province. National Statistics Institute (INE) [47], and Dwellings with Foreclosure Begun by Province. National Statistics Institute (INE) [48]. Author's elaboration.

According to Ian Brossat [49], this new platform capitalism represents an advance that even the best dreams of neoliberalism 20 years ago did not foreshadow, since, although it began by putting people who needed accommodation in contact with others who could provide it, it included economic exchange and turned collaboration into transaction, externalizing costs and risks and basing this economic activity on deregulation.

The above processes have been recognized in the whole of southern Europe, where critical urban scholarship has made large use of gentrification, touristification, and financialization to explain the impacts of crisis, austerity, and subsequent economic rebound driven by the real estate and tourist sectors [50].

### 3.3.1. Which Signs of Urban Restructuring Characterize These Moments of Repossession?

Signs of urban restructuring include a process of strengthening tertiary specialization, urban recentralization, and residential and tourist gentrification.

The analysis of the territorial dynamics in Spain based on the microdata of the Residential Variation Statistics carried out by Otero et al. [51] showed that the cities have recovered a clear leadership in the post-crisis stage. Therefore, from the point of view of flows, they have been much more attractive than peri-urban, ex-urban and rural areas. That is to say, the suburban and counter-urban scenarios, and the revitalization of the so-called empty or emptied Spain, which acquired a certain prominence during the period of prosperity, have shown great volatility. Therefore, as opposed to the years of suburbanization and transformation of metropolitan areas, we are today witnessing a stage of "return to the compact city" or, at least, this was the case until the pandemic appeared. Hence, our gaze is

directed at the dynamics, tensions, and conflicts that are taking place in the central spaces of Spanish cities.

In recent years, the process of residential elitization that has affected urban centers has been accompanied by an increase in dispossession and repossession. If in the toughest years of the crisis, evictions for unpaid mortgage payments were concentrated in areas of recent urbanization, in the residential peripheries of most cities, evictions for non-payment of rent have been common in urban centers. According to Domínguez et al. [52], although it is true that the development of tourist housing is not a direct factor in eviction in general terms, there is a territorial relationship between the presence of this form of tourism and the loss of property ownership and use.

There is no doubt, therefore, that the tourist gentrification that was activated in the post-economic crisis is impacting on the housing market of the compact city, given the increase in rent and the price of property owned, which makes it inaccessible to the lower income segments of the population to purchase or rent. In addition to these changes related to housing, there are also transformations in commerce, and a phenomenon of urban artificialization, due to the effects of tourism.

Cities tend to offer an image of homogeneity, which is identified with a place for tourist visits, with the aim of satisfying the interests of those visitors, while at the same time consolidating itself as a space-museum in which it is possible to enjoy what makes it peculiar or different in the eyes of the tourist. Thus, for example, the central supply markets are losing customers to the supermarkets, so they are reinvented as gourmet spaces, while at the same time being the object of guided tours with the sole purpose of providing tourists with a photograph of the local products that they take with their mobile phones.

In this scenario, it should be remembered that the ex novo tourist developments gave rise to urban tourist forms that were different from urban entities. However, the transformative potential that tourism is exercising today in consolidated cities makes the boundary between tourist and urban spaces much more blurred, as well as the very conception of what we understand by the term "city".

Finally, these trends of urban restructuring in recent times, in line with the so-called third globalization, are causing a great challenge from the perspective of scale, given that the forces of internationalization of real estate capital, whether of investment companies, the fortunes of individuals mobilized in the search for niches of stability, or online business platforms, are contrasted with a municipal management that has to resolve the local interests of multiple actors: neighbors, small traders, owners of conventional tourist establishments, etc., which is a difficult undertaking for public servants when they want to look after the interests of citizens.

### 3.3.2. Which Social Transformations Are Taking Place in This Incipient Post-Crisis Period? Are We Capable of Recognizing Them?

The right to the city, defined by Henri Lefebvre [53] as the right of urban dwellers to build, decide, and create the city, and to make it a privileged space of anti-capitalist struggle, loses force in times of what Ulrich Beck defined as the risk society [54] and Zygmunt Bauman called liquid modernity [55], that is, in moments of vulnerability as well as of fluidity, hybridization and flexibility. Could the post-crisis urban society be defined as a liquid citizenship? Is this interpretation sufficient to describe the society of the Late-Capitalist City or of the third globalization?

I believe that the answers to these questions are not simple, because as I pointed out earlier, while we are moving toward devaluation and revaluation, we are also moving toward equalization and differentiation. There is no doubt that we are living in times of temporariness, to which the deterioration of the labor market, the loss of employment or increase in unemployment, the precariousness of work and temporariness, the spread of discontinuous labor trajectories, marked by chronic insecurity, low income, and the absence of prospects for improvement have definitely contributed.

This is compounded by the wage devaluation caused by the labor market reform of 2012 [56], which leads to a lack of social protection, a situation of vulnerability that is inherent in the neoliberal mode of regulation. However, citizens are also anxious about novelties and incessant changes, and so they see tourists multiplying in cities, becoming themselves urban tourists from other cities; grouping themselves according to diverse interests and demands, of identity, gender, lifestyles; and coexisting in societies where mobility and immigration are resumed, diversified, and intensified, until the moment of the pandemic.

The capacity for adaptation and flexibility in the face of new situations also hides resistance and acts of demonstration, promoted by protest platforms that fight for the right to housing, demanding legislation for holiday homes, for the construction of social housing, for the regulation of the rental market, and for the penalization of owners of empty houses.

The interests of the different platforms are mixed in such a way that those who lead the demands for the right to housing, also demand improvements in pensions, another of the distinctive signs of the city of our times, the city of demographic aging; or they form part of groups that demand urban measures to make cities more sustainable, from the point of view of greenhouse gas emissions or of mobility.

It can be said that the result of all this is complexity and lack of definition, since this capacity to respond, this collective will to shape the city, is affected by the resistance of capital and its spokespersons, making it increasingly difficult to develop the necessary political strategy to invoke dignity through a pact between humanity and technology, in what have come to be called the human-centered smart cities [57].

As Méndez pointed out [58], the great challenge of the present will involve promoting structural change by building cities that are more intelligent in terms of their economic base, their socio-labor structure, and their public management; but also, cities that are more livable by reducing problems of exclusion, improving the quality of the built space, and reducing their ecological footprint. Only in this way, according to the Global Platform for the Right to the City [59], will we citizens become actors building a more dignified city together.

## 4. Conclusions. The Uncertain Future of the Spanish City in Times of Pandemic

The reflections that guided this study were affected by an unexpected pandemic expansion that has modified our perception of the world and our habits of life, and whose urban impact we are still unable to evaluate. In this context we must express the following questions: Have cities and their neighborhoods been punished with the same epidemic and socioeconomic intensity? What will the Spanish cities' next generation be like in an era where physical contact between people is restricted? Will there be notable changes in the socioeconomic structure that guides urban evolution? I believe we are in a position to answer these questions negatively, as the images of the so-called "hunger queues" speak for themselves. This was already expressed by Ana Fani A. Carlos [60] when she said that the virus is deepening the social crisis in an unequal way. This unequal dimension is related not only to the contagion, but also to the economic, social, political, and institutional substrate that in some places accentuated the destructive capacity of the pandemic and its consequences, while in others it found greater defenses against its impacts [61].

There is no doubt that COVID-19 has reached planetary dimensions because, in a globalized world, preventing transmission seems impossible. For this reason, we must reflect on environmental conditions, density factors, the age composition of the population, mobility, productive structures, and city government, but we are still a long way from providing a reasonable interpretation of these questions. However, despite this lack of knowledge and the widespread crisis that the pandemic has caused, there are some positive changes, which should be highlighted. From a socioeconomic point of view, it is worth mentioning the generalization in the use of information and communication technologies. The distance working model, distance training, and online commerce have been reinforced in addition to other services, such as the health service itself, which has also adopted

new forms of provision. Business digitalization has increased, and the production and marketing activities of certain local businesses have been strengthened, particularly in the agro-food sector.

From a socio-political point of view, there has been empowerment of the public sector, given that there has been a growing control in the management of daily life by central, regional, and local government authorities. At the same time, cooperation has been consolidated, as demonstrated by the agreements reached within the European Union, health coordination initiatives between autonomous communities and other urban governance actions. From an environmental point of view: gas emissions and pollution have been reduced; proposals to promote sustainable tourism have multiplied and local tourism has been strengthened; but there has been a decrease in the use of public transport, with some revitalization of the private car and of bicycles and skateboards.

From an urban point of view, there is a new tendency to oppose the strategies to rein-force compact cities. These last were defended before the pandemic, trying to avoid urban expansion and land occupation in the peripheries. Nonetheless, in the current times most citizens show a preference for suburban areas and small urban centers identified as being safer spaces. Finally, from the point of view of housing, some legislative initiatives have been developed to curb the tendency toward foreclosures and evictions in times of unemployment and social vulnerability.

The above transformations characterize these times of pandemic, but it is difficult to predict whether they will continue in the future. I am confident that, on the post-pandemic horizon, the positive circumstances can become an opportunity to face the challenges in the transition of Spanish cities toward intelligent urban planning models, especially regarding energy and environmental issues. Nevertheless, we must not forget the disparate inequality of the pandemic effects, geographical evidence that we will have to study in depth, following the advice of Iván Serrano et al. [62] when they paraphrased Chesterton [63], pointing out that geography has, more than ever, the task of thinking with a perspective beyond immediacy and drawing up a common agenda in which we consider what's wrong with the world.

**Funding:** This work was supported by the European Regional Development Fund (ERDF)/the Ministry of Science, Innovation and Universities–State Research Agency (AEI) under R&D Project "Housing and international mobility in cities of the Canary Islands. The emergence of new forms of urban inequality" (RTI2018-093296-B-C21).

**Institutional Review Board Statement:** Not applicable.

**Informed Consent Statement:** Not applicable.

**Data Availability Statement:** Raw data are publicly available.

**Conflicts of Interest:** The author declares no conflict of interest.

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
