# Peer review of "The Urban Mirror of the Socioeconomic Transformations in Spain"

_urbansci, doi:10.3390/urbansci5010013_

Round 1

Reviewer 1 Report

I found the manuscript interesting and worth of publication. I mostly agree on all the points raised in the article. This is a good contribution to urban science. I have only one concern, depending on the presentation form. I see here the classical article organization with introduction, methodology, results, etc. I see it a bit artificious, considering that the article is not reporting a specific methodology or results of quantitative/qualitative ad-hoc analysis. My suggestion is to turn presentation to a 'commentary' which is perfectly in line with the content of the study. Doing so, the author has some more degrees of freedom to add further references to the Spanish context and the similar international contexts in Portugal, Italy, Greece, etc., doing a more complete vision of recent patterns of Mediterranean urbanization. Thank you.

Author Response

To Reviewer 1.

  1. I agree to turn the document under examination as commentary given that the article is not reporting a specific methodology or results of quantitative/qualitative ad-hoc analysis. Therefore, the document has been proposed to Urban Science in the section of Commentary instead of the section of Article.
  2. Further references to the Southern European context have been added, as suggested. Specifically, these are in Section 3.1. “Furthermore, it remained a common trend in Southern Europe, where over-urbanization based on hyper-compact and deregulated expansion, real estate speculation and informal economy were general factors of the cities’ growth” In Section 3.2. “In Southern Europe, austerity was marked by a shift towards a starker version of neo-liberal doctrine, but the crisis did not lead to the reforms expected, at least not in the field of housing” In Section 3.3. “The above processes have been recognised for the whole of Southern Europe, where critical urban scholarship has made large use of gentrification, touristification and financialisation to explain the impacts of crisis, austerity and afterwards economic rebound driven by real estate and tourism”.

Reviewer 2 Report

This study well presents the economic and social transformations of Spanish cities in recent years by taking an in-depth literature review from the perspectives of economic expansion, economic crisis, and the rise of uncertainty. The author thoroughly delivers the dimensions of the linkage between socio-economic dynamism and its spatio-temporal consequences on urban fabric, and effectively develops the insights on how we prepare for the advent of "New Normal". Elaborating those lessons and ideas with the Spanish case would be a right choice since Spain has been experiencing fierce dynamics from the early wealth expansion to 2008-2014 financial crisis -- and now to the COVID-19 pandemic. I believe that the current version of the manuscript would acquire the “ready-to-be-published” status, after considering the following minor points.

(1) From the line 387 to 397, I think the emergence of new residential technologies, such as Airbnb, would not directly affect the existing housing market or the impact would be minimal, at least in the short term. Adding new options for residing people might be an opportunity for the expansion in the housing sector. Rather, this lodging industry might be at stake due to the recent pandemic.

(2) Although this study mostly focuses on the “reflections”, I want to encourage you to provide your insights on what the next-generation Spanish cities would or should be in this era where physical contacts between people are restricted. How should urban spatial structure evolve to support socio-economics activities in the upcoming years -- in the Spanish context? I hope your perspectives or forecasts be added to the Conclusion section.

Author Response

To Reviewer 2.

2.1. I tried to deepen in the nexus between the processes of foreclosure and the touristification effects, showing their non-parametric correlation (Spearman).

2.2. I have attempted to provide my insights into what the next generation of Spanish cities would be like in post-pandemic times. These perspectives or forecasts have been added to the Conclusion section, renamed as “The uncertain future of the Spanish city in times of pandemic”.

Reviewer 3 Report

This is an interesting topic relating between economic expansion, economic crisis and sharing economy to urban configurations and gentrification. However, the relationships are not clearly demonstrated with evidence. It requires a major revision and the following are the comments.

  • Abstract

It summarizes the socioeconomic transformation stages, but it does not address how do these stages shape the city characteristics and why. It does not state the findings of the study.

  • Introduction

The statement of the differences between cities is important, but the explanation of the “two forces” (quoted below) is not comprehensible.

“…the result of the interaction of two forces: the concentration of capital, labour and culture in the city and the radical transformation of its economic base, with the passage from a Keynesian and Fordist system of mass production and consumption to a post-Fordist system of flexible, information-intensive industrialisation associated with the vertical disintegration of the production process.”

The research question and the aims of the study are not clearly spelt out.

  • Materials and Methods

It does not provide enough information on the materials and methods used in the study. For example, how are the publications selected? How many publications are selected? Why are they selected? What is their chronological pattern? What is the methodology employed to demonstrate with evidence the relationship between the socioeconomic events and urban changes?

  • Results

Economic expansion stage – the association between economic expansion and urban configuration is not clearly explained with evidence. For example, a city size growing map associated with an economic expansion map may help indicate how the two are related.

Economic crisis stage – the association between economic crisis and urban crisis is also not clear and no evidence is provided. For example, a map showing the vacant houses before and after the crisis map help illustrate how the economic crisis shaping the urban crisis.

Uncertain future – The map showing the internationalization of housing (Figure 3) shall show the changes before and after the crisis, so as to demonstrate the increasing % of foreign buyers. Furthermore, what are the impacts of internationalization of housing shall be elaborated. Is it higher housing prices or gentrification or other urban changes? If it is about housing prices, then a chart showing the association between housing prices and the international housing buyer percentage shall be provided to demonstrate the impact. The second point on collaborative economy of real estate shall also provide a similar map indicating the association between Airbnb and gentrification.

  • Concluding Remarks

It does not make any concluding remarks of the study, but discusses the impacts of the pandemic on city inequality.

Author Response

To Reviewer 3.

3.1. Abstract. A shallow mention on how these stages shape the city characteristics has been added, respecting the length fixed by the Urban Science instructions for authors.

3.2. Introduction. The sentence about the “two forces” has been clarified.

3.3. Materials and methods. The section has been re-elaborated attending to the suggestion of the reviewer.

3.4. Results:

  1. a) Economic expansion stage. Attending to the suggestion of the reviewer one map has been added and commented. Figure 1. Variation of artificial land (2000-2006) %.
  2. b) Economic crisis stage. Attending to the suggestion of the reviewer one map has been added and commented. Figure 4. Difference in the number of Housing Properties Transferred in 2013 compared to 2008.
  3. c) The capitalist restructuring of the Spanish city after the economic crisis. Attending to the suggestion of the reviewer a chart on the evolution by quarter of the housing prices and of the housing acquisitions by foreigners between 2014 and 2019 has been added and commented (Figure 6). And a combined map: Figure 7. Number of Dwellings with begun foreclosure by province, 2014-2019; and Distribution of the Tourist Dwellings by province, in absolute figures in 2020.

3.5. Concluding Remarks.

As this section does not make any concluding remarks, it has been renamed as: The uncertain future of the Spanish city in times of pandemic.

Round 2

Reviewer 3 Report

Abstract

  • What are the methodology and the findings of this study?

Introduction

  • What are the research question(s) and the aims of the study?

Materials and Methods

  • It adds information on the materials referred, but not the methods used in the study. What is the methodology employed to demonstrate with evidence the relationship between the socioeconomic events and urban changes?

Results

  • Economic expansion stage – In Figure 1, it would be good to show the GDP growth from 2000 to 2006 of the provinces as an evidence of the association between economic expansion and artificial land expansion.
  • Economic crisis stage – Figure 4 shows the reduction in house transfers after the crisis. It is not a good indicator of a decrease in housing demand. Either a map showing the vacancy rate changes, or a map showing house price falls after the crisis is more relevant.
  • Uncertain future – The map showing the internationalization of housing (Figure 3) shall show the changes before and after the crisis, so as to demonstrate the increasing % of foreign buyers.
  • Internationalization effect on house prices – Figure 6 seems to reject your claim of their association. For example, if we compare between 2014Q1 and 2019Q2 (both ends of your figure), when house price increased from about 9000 to 16000, i.e. about more than 75% rise, but the housing acquisitions by foreigners just increased from 1250 to 1350, i.e. about 8% increase. How can an 8% increase in foreign buyers causes a 75% increase in house prices? It requires more justifications.

Conclusion – no conclusion section

Author Response

Comments to Reviewer 3

Abstract, Introduction, Materials, Methods and Conclusions

The raison d'etre of the study developed does not correspond to an article. It has been classified as Commentary and this type of document does not require the same structure and format. This has been mentioned in the abstract where can be read: “An in-depth bibliographical review leads to this commentary”.

Results

Figure 1 has been reworked adding the information of the Interannual Variation Rate of the GDP for the same period.

Figure 4 has been modified. By the one hand, this figure now shows the negative difference in the general housing price index (Base index 2015) between 20081Q and 20141Q by Autonomous Communities, given that there is not information at the level of provinces. By the other hand, there is not information about vacant houses in Spain beyond the data provided by the 2011 Census.

Figure 5 has not been modified given that the aim of this map is to show the importance of housing investors in the Mediterranean provinces and the two archipelagos, in relative terms, in 2019, as indicated in the text.

A clarification regarding figure 6 has been done. With this clarification the idea of the role of foreign investment in the increase in housing prices has been relativized.

Round 3

Reviewer 3 Report

Thanks for the clarifications and revisions. I have the following minor comments for further improvements.

  1. It is good to add GDP ranges in Figure 1, but it requires some explanations on how do they reflect the argument. The variation of artificial land does not seem to be positively correlated with the variation of the GDP, why the urban expansion is said to be associated with economic expansion?
  2. It is good to use house price index in Figure 4 to illustrate the impact of the financial crisis on housing markets. However, it is confusing by using "Negative difference". When the title of the figure is Difference in ..., but the legend shows "Negative difference ...", can they be aligned? What is Negative difference? If it means light blue colored area was 0 to 40 points decrease compared to 2015 index, then it seems that it is not the Southern Spain that encountered the heaviest blow, but the northern Spain. It requires some explanations on how the map supports the argument.

Author Response

Responses to reviewer 3:

The the first question:

This comment to figure 1 has been added.

The increase in artificial land in Spain can be recognised on the map in Figure 1, which represents its variation between 2000 and 2006, according to data from the European CORINE (Coordination of Information on the Environment) Land Cover project. The GDP Interannual Variation Rate (%) has also been depicted in the figure. But there is not a clear positive correlation between these indicators by provinces, given that the artificial land’s increase is also related to the investment in second houses, to foreign investment and to a process of suburbanization that sometimes exceeds the provincial limits.

The second question:

The map has been modified eliminating the word negative.

The comment related to the map has been modified clarifying the main reason for the regional disparities in the fall of housing prices according to the studies of García Montalvo.

"A good example of the drop in the general housing price, by Autonomous Communi-ties, in 20081Q-20141Q is shown in Figure 4. This dynamic is related to the fact that in those Autonomous Communities where the prices were highest at the beginning of the real estate bubble bust, the fall was greater [26]."

[26] García Montalvo, J. (2013). Dimensiones regionales del ajuste inmobiliario en España. La economía de las regiones españolas en la crisis, Papeles de Economía Española, 138, 2013, 62-79.
